# Long-term outcome of tibial plateau leveling osteotomy using an antimicrobial silver-based coated plate in dogs

**Geoffrey Pagès**⊙*, **Meike Hammer, Jean-Guillaume Grand, Iban Irubetagoyena**

Surgery Department, Centre Hospitalier Vétérinaire Aquivet, Eysines, France

* geoffrey.pages@gmail.com

## Abstract

### Objectives

To evaluate long-term outcome using the BioMedtrix™ TPLO Curve® plate in dogs with cranial cruciate ligament disease (CrCLd) treated by tibial plateau leveling osteotomy (TPLO).

### Study design

Retrospective case study.

### Animals

Dogs (n = 323, 337 stifles).

### Methods

Medical records were searched for dogs presented with CrCLd and treated by TPLO with the BioMedtrix™ TPLO Curve® plate for 3.5 mm screws between March 2018 and December 2020. Tibial plateau angles (TPA) were measured on preoperative, postoperative, and follow-up radiographs. Changes in TPA between postoperative and follow-up radiographs (ModTPA) were calculated. Radiographic bone healing was scored. Complications were evaluated. Surgical site infections (SSI) were compared to a control group of dogs treated with the Synthes™ TPLO plate between January and December 2017. Owners of both groups were contacted by telephone at least 1 year postoperatively.

### Results

The BioMedtrix™ group was composed of 237 dogs (248 stifles), the control group was composed of 86 dogs (89 stifles). In the BioMedtrix™ group, radiographic follow-up was performed at a median of 48 days. Average ModTPA was 1.2˚. Bone healing was graded as complete, good, poor, and none in 18%, 62%, 20%, and 0%, respectively. At a median of 786 days postoperatively, minor and major postoperative complications were observed in 6 (2.4%) and 32 (12.9%) cases in this group, respectively. During the first year following the surgery, 23 (9.3%) and 12 (13.5%) cases suffered a SSI, of which 12 (4.8%) and 7 (7.9%) had their implant explanted in the BioMedtrix™ group and the control group, respectively.

**Data Availability Statement:** Data relevant to this study are available from figshare at 10.6084/m9.figshare.20379918 (https://doi.org/10.6084/m9.figshare.20379918.v1) and 10.6084/m9.figshare.

20379921 (https://doi.org/10.6084/m9.figshare.20379921.v1).

**Funding:** The author(s) received no specific funding for this work.

**Competing interests:** The authors have declared that no competing interests exist.

There was no significant difference between groups for SSI and implant explantations (p = 0.31 and p = 0.29, respectively).

## Conclusion

The BioMedtrix™ TPLO Curve® plate provided a reliable fixation system for osteotomy healing after TPLO. Bone healing and long-term complication rates were similar to previous studies using other implants. SSI rates were similar between the BioMedtrix™ group and the control group. The antimicrobial HyProtect® coating of the plate did not reduce SSI in this study.

## Clinical significance

The BioMedtrix™ TPLO Curve® plate can be safely used for TPLO. The value of the antimicrobial HyProtect® coating of the plate may be questioned, as SSI rate was not lower in this study compared to the control group or previous reports.

## Introduction

Cranial Cruciate Ligament disease (CrCLd) is one of the most common orthopedic conditions in dogs [1].

CrCLd is described as a multifactorial degenerative disease, involving genetics, morphology (tibial plateau angle (TPA), distal femoral anatomy), chemical factors, and immunity [2].

Surgical management is the recommended treatment option. Currently, the tibial plateau leveling osteotomy (TPLO) described by Slocum in 1993 offers a reliable outcome and is one of the most practiced techniques to treat CrCLd in dogs [3,4].

TPLO was initially described using non-locking plates [3]. Several modifications have been implemented to improve fixation, screw orientation, and plate fit to the proximal tibial anatomy. Studies comparing TPLO plates concluded that there is improved osteotomy site stability using locking plates [5–8]. In previous studies, TPA modification between postoperative and follow-up radiographs (ModTPA) or bone healing grading have been used to assess the construct stability [7–9].

Reported complication rates after TPLO have been reduced over the past decades, due to improvements in surgical technique and implant refinements. In recent large-scale studies, postoperative complication rates ranged from 11.4% to 14.8% [10,11]. Surgical site infection (SSI) is the most common complication, ranging from 3.6% to 16.5% [9–12]. Several modifications to perioperative antibiotic prophylaxis, postoperative antibiotic treatment, or postoperative wound dressings have been proposed in an effort to reduce SSI [13–15].

A new TPLO plate was recently released by BioMedtrix™, designed for improved fit to conform to the shape of the proximal tibia. It further includes a polyaxial locking system, using a system-specific drill guide which allows screw angulation of up to 12.5˚ to reduce the incidence of intra-articular screw placement. Finally, the plate incorporates an antimicrobial silver-based coating (HyProtect™), described by the manufacturer as providing protection from local infection for at least 100 days.

The main objective of this study was to evaluate the outcome (radiographic evaluation and long-term complications) of dogs treated with the BioMedtrix™ TPLO Curve® plate, which has never been published to the authors' knowledge. The second objective of this study was to

focus on SSI rate during the first year after surgery. We hypothesized that at the time of radiographic follow-up: (1) the ModTPA and (2) bone healing rates would be similar to previous reports. We also hypothesized that (3) long-term postoperative complications, and particularly, (4) SSI rates would be lower compared to a control group and previous studies reporting the use of non-coated implants.

## Materials and methods

### Study design

This study followed international guidelines for humane animal treatment and complied with legislation (EU Convention on the protection of animals revised directive 86/609/EEC).

Dogs were included in the study group (BioMedtrix™ group) if CrCLd was diagnosed and treated by TPLO using the BioMedtrix™ TPLO Curve® plate for 3.5 mm screws, between March 2018 and December 2020, by two surgeons (Dipl. ECVS) of a single referral center, and if the recheck and follow-up radiographic examination was performed by the primary surgeon. A control group was formed with dogs treated by TPLO using the Synthes™ TPLO plate for 3.5 mm screws, between January and December 2017, by the same two surgeons. Follow-up radiographic examination wasn't an inclusion criteria for this group.

Dogs with a history of stifle surgery on the affected limb, presented with potential infectious condition (local or generalized dermatitis, prostatitis. . .), or requiring an additional stifle joint procedure (patellar stabilization. . .) were excluded. Additional procedures due to complications (septic arthritis, implant explantation, late meniscal tear. . .) were not cause for exclusion.

Data obtained from the medical records included signalment (breed, age, gender, weight), preoperative radiographs, side of the lesion, type of cruciate ligaments lesions, and type of implant used. For the BioMedtrix™ group, data obtained also included postoperative radiographs, recheck radiographs, and complications. For the control group, additional data only consisted of the occurrence of a SSI and potentially associated implant explantation.

### Surgical procedure

Pre-anesthetic blood tests were performed depending on the animal's condition. The anesthetic protocol was established at the surgeon's discretion. Premedication included either diazepam (Valium; Roche™, Boulogne-Billancourt, France), 0.25 mg/kg intravenously (IV) or acepromazine (Calmivet; Vetoquinol™, Lure, France), 0.05 mg/kg intramuscularly (IM) and morphine (Morphine; Lavoisier™, Paris, France), 0.2 mg/kg IV. Induction was performed with propofol (PropoVet; Zoetis™, Malakoff, France) 4 mg/kg IV, to effect. Anesthesia was maintained with isoflurane (Isoflurin; Axience™, Pantin, France), 1.5–2.5%, in 100% oxygen. Preoperative orthogonal radiographs of the affected stifle were performed under general anesthesia. Antibiotic therapy consisted of cefazolin (Cefazoline; Mylan™, Saint-Priest, France), 22 mg/kg IV 30 minutes before incision, and was repeated every 90 minutes until skin closure. Articular exploration was performed by either arthroscopy or medial « mini-arthrotomy » [16], and meniscal tears were treated by partial meniscectomy, hemimeniscectomy, or total meniscectomy as indicated. Approach to the proximal tibial metaphysis and TPLO were performed as described by Slocum [3,17]. The plate was placed as proximally as possible. Contouring of the plate was done at the discretion of the surgeon. To avoid intra-articular screw placement in some cases, the most proximal screw was angled in a medio-proximal to latero-distal direction using the specially designed conic threaded drill guide. All proximal screws were locking screws and all distal screws were standard cortical screws. Compression of the osteotomy was achieved, first, using bone holding forceps and, second, by placing the first and third distal screws in compression. Finally, tightness of all screws, range of movement, limb alignment,

patellar stability, and tibial compression test were assessed. The incision was closed routinely using polydioxanone (Monotime; Péters Surgical™, Bobigny, France), and polyamide monofilament (Filapeau; Péters Surgical™, Bobigny, France). A sterile wound dressing was applied to the wound (Hydrofilm Plus; Hartmann™, Châtenois, France). Postoperative orthogonal radiographs were performed under general anesthesia. A modified Robert Jones bandage was applied for 24 hours to limit wound swelling at the surgeon's discretion. Postoperatively, dogs were medicated with morphine (Morphine; Lavoisier™, Paris, France), 0.2 mg/kg subcutaneously every 4 hours, and cefalexin (Rilexine; Virbac™, Carros, France), 20 mg/kg orally twice daily, until discharge. Local cryotherapy was applied every 8 hours during hospitalization. Dogs were discharged 24 to 48 hours after surgery with cefalexin (Rilexine; Virbac™, Carros, France), 20 mg/kg orally twice daily for 5 days, and meloxicam (Metacam; Boehringer Ingelheim™, Reims, France), 0.1 mg/kg orally once daily for 10 days. Sutures were removed 15 days after surgery.

Short leash walks were the only recommended activity 3 to 4 times daily until radiographic recheck.

## Radiographic evaluation

Two observers (1 Dipl. ECVS and 1 ECVS resident) assessed the anonymized radiographs on a DICOM viewer (Horos™ v3.3.6, horosproject.org). All observers were experienced in TPA measurement (more than 3 years) [3]. Preoperative TPA (PreTPA), postoperative TPA (PostTPA) and TPA at recheck (ReTPA) were measured as previously described [18]. ReTPA minus PostTPA (ModTPA) was then calculated. Bone healing was graded using a scale developed by Oxley and modified to allow numerical grading (Table 1) [12]. Owners of dogs presenting insufficient bone healing (grades 0 and 1) were asked for a recheck after 4 to 6 weeks to ensure sufficient bone healing (grades 2 or 3).

## Complications evaluation

Complications were categorized as minor, major, and catastrophic, as previously described by Cook *et al.* [19]:

- Catastrophic: complication or associated morbidity that causes permanent unacceptable function, is directly related to death, or is cause for euthanasia,

- Major: complication or associated morbidity that requires further treatment based on current standards of care:

  - Requires surgical treatment to resolve based on current standard of care,

  - Requires medical treatment to resolve based on current standard of care.

- Minor: not requiring additional surgical or medical treatment to resolve (eg, bruising, seroma, minor incision problems, etc.).

**Table 1. Radiographic bone healing scale.**

| Score | Healing grade | Description |
|---|---|---|
| 0 | None | No biological activity |
| 1 | Poor | Some evidence of bone healing but unbridged cortices and/or osteotomy gap |
| 2 | Good | Active bridging callus and/or osteotomy mostly blurred or filled with callus |
| 3 | Complete | Remodeled callus at all cortices and/or osteotomy indistinct |

Specifically, SSI were diagnosed if at least one of the following abnormalities was observed [20]:

- Purulent drainage from the incision, with or without laboratory confirmation,

- Organisms isolated from an aseptically obtained culture of fluid or tissue from the incision,

- An abscess or other evidence of infection involving the incision found on direct examination, during reoperation, or by histopathologic or radiologic examination,

- Diagnosis of a SSI by a surgeon or attending physician.

Beyond the first postoperative year, medical records were searched for long-term complications, particularly any information related to a SSI, and the owners were contacted by telephone and questioned about complications during the first postoperative year.

## Data analysis

Data were analyzed using a statistical software (R 4.0.2, The R Foundation for Statistical Computing, Vienna, Austria). Statistical analysis for categorical data used either Chi-squared (breeds) or Fisher's exact (sex, CrCL side, type of CrCL lesion, SSI, implant explantations) tests. For continuous data, after normality evaluation using Shapiro-Wilk test, analysis used either Student's t (age, weight, PreTPA) or Mann-Whitney-Wilcoxon (ModTPA, bone healing) tests. Interobserver variability was estimated by the standard deviation of the observer's mean measurements of TPA per dog. Post hoc power analysis was performed to evaluate potential differences between groups for SSI and implant explantation rates, using $\alpha = 0.05$. Statistical significance was set at $p < 0.05$.

## Results

### Descriptive analysis

Out of 415 dogs initially identified, 323 dogs (337 stifles) met all inclusion criteria The Bio-Medtrix™ group was composed of 237 dogs (248 stifles), the control group was composed of 86 dogs (89 stifles).

In the BioMedtrix™ group, the most commonly represented breeds included Labrador Retriever (n = 43), Golden Retriever (n = 27), Cane Corso (n = 21), German Shepherd (n = 13), and Boxer (n = 12). The median age was 5 years and 5 months old (0.5–12 years old). There were 137 (55.2%) females (73.7% of which were spayed) and 111 (44.8%) males (35.1% of which were neutered). The median weight was 34.2 kg (16.4–80 kg). One hundred and thirty-six left and 112 right stifles were included. One hundred and sixty-three (66.0%) cases presented a complete CrCL rupture, and 84 (34.0%) cases presented a partial CrCL tear. Fifteen (6.1%) cases presented a caudal cruciate ligament partial tear. Median radiographic follow-up was 48 days (±12 days, 30–104 days). The long-term follow-up via telephone was available for 192 (77.4%) cases, at a median of 786 days after the surgery (±90 days, 642–923 days).

In the control group (Synthes™ group), the most commonly represented breeds included Labrador Retriever (n = 19), Golden Retriever (n = 12), Boxer (n = 8), Cane Corso (n = 7), and German Shepherd (n = 6). The median age was 5 years and 1 month old (1.5–11 years old). There were 46 (51.7%) females (82.6% of which were spayed) and 43 (48.3%) males (41.9% of which were neutered). The median weight was 32.4 kg (18.1–56 kg). Forty-two left and 47 right stifles were included. Type of CrCL lesion was known for 78 cases, including 65 (83.3%) cases presenting a complete CrCL rupture, and 13 (16.7%) cases presenting a partial CrCL

tear. Caudal cruciate ligament status was known for 53 cases. Five (9.4%) cases presented a caudal cruciate ligament partial tear. The long-term follow-up via telephone was available for 67 (75.3%) dogs at a median of 1145 days after the surgery (±102 days, 995–1293 days).

Breeds, age, gender, body weight, and CrCL side were similar between the BioMedtrix™ group and the control group (p = 0.87, p = 0.77, p = 0.62, p = 0.65, and p = 0.22, respectively). The type of CrCL lesion was significantly different between groups (p = 0.004).

## Radiographic evaluation

Mean PreTPA was 25.5˚ (±4.1˚, 14.2–40.5˚). Mean PostTPA was 3.7˚ (±3.1˚, -8.7–13.7˚). Mean ReTPA was 4.5˚ (±3.3˚, -7-14˚). Mean ModTPA was 1.2˚ (±1.1˚, 0.0–9.4˚). Measured PreTPA (mean, SD) were significantly different (p = 0.028) between observer 1 (24.8˚, ±4.42˚) and observer 2 (26.3˚, ±4.6˚) (Fig 1). Measured PostTPA, ReTPA, and ModTPA (Fig 2) were not significantly different between observers (p = 0.51, p = 0.95, p = 0.065, respectively).

Mean bone healing was graded as complete (grade 3), good (grade 2), poor (grade 1), and none (grade 0) in 18%, 62%, 20%, and 0% cases, respectively (Fig 3). Measured bone healing was not significantly different between observers (p = 0.92).

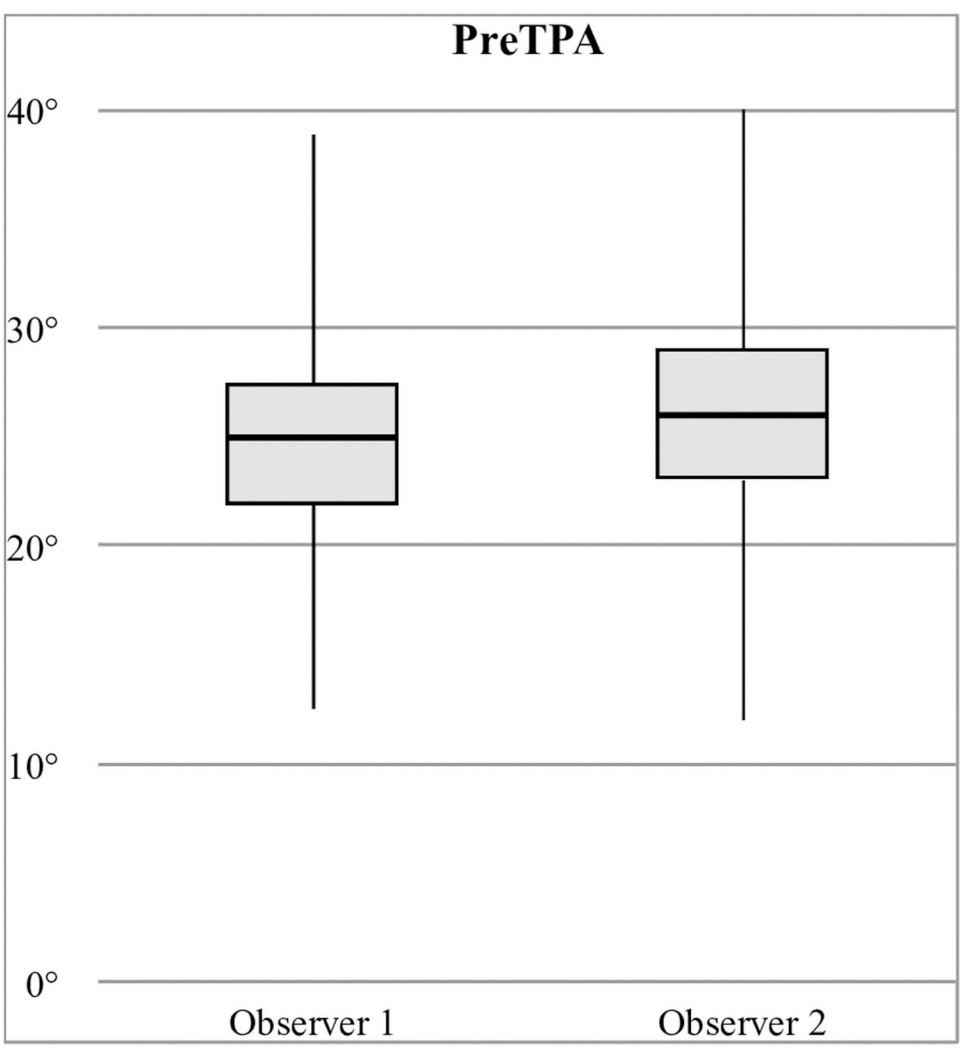

**Fig 1. PreTPA differences in radiographic measurements between observers.**

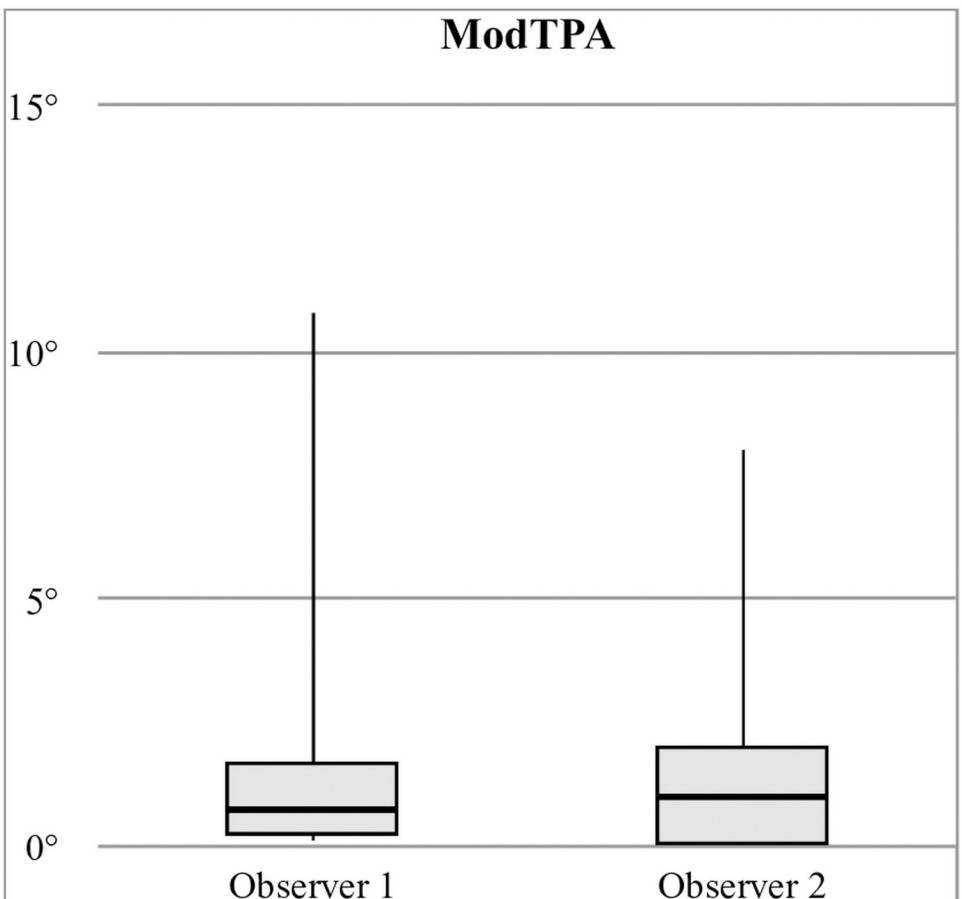

**Fig 2. ModTPA differences in radiographic measurements between observers.**

### Intraoperative complications

Fifteen (6.0%) intraoperative complications were reported, including 6 (2.4%) minor, 9 (3.6%) major, and no catastrophic complications.

All minor intraoperative complications were fibular fractures, which developed during tibial plateau rotation.

All major intraoperative complications were hemorrhage of the cranial tibial artery. Hemorrhages were controlled with either direct pressure over the bleeding site with sterile gauze or application of vascular clips through the osteotomy gap, as described by Matres-Lorenzo [21].

No catastrophic intraoperative complication was reported. In particular, no intra-articular screw placement was reported.

### Postoperative complications

Thirty-eight (15.3%) postoperative complications were reported, including 6 (2.4%) minor, 32 (12.9%) major and no catastrophic complication (Table 2). In total, 37 (14.9%) dogs suffered postoperative complications.

Minor postoperative complications included clinical patellar tendinitis with associated discomfort on palpation (n = 2), fibular fracture (n = 1), implant failure (n = 1), patellar fracture (n = 1), and tibial crest fracture (n = 1). Both patellar tendinitis were judged mild, and no

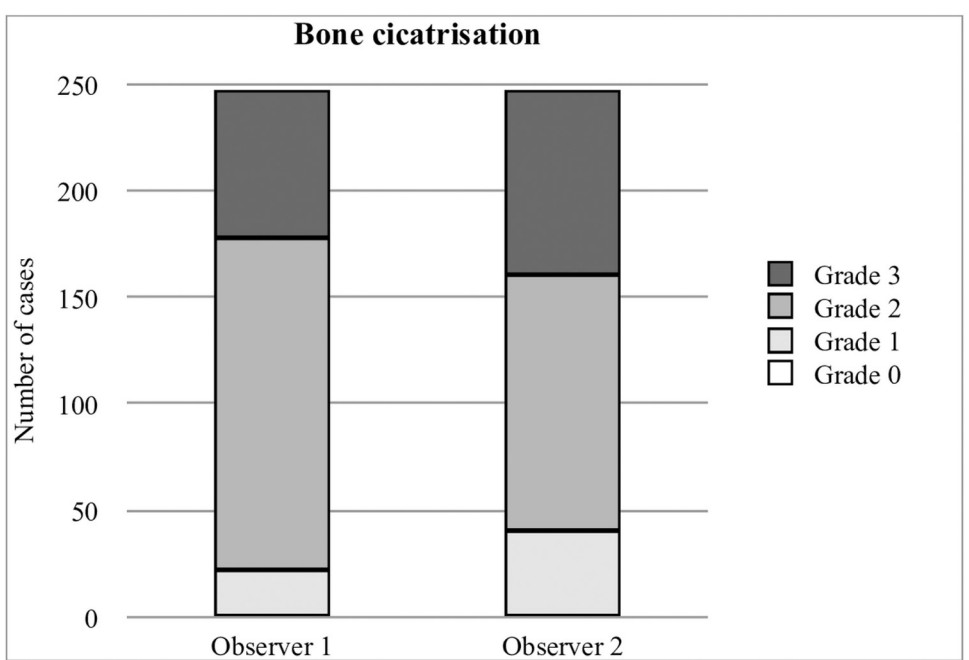

**Fig 3. Differences in bone healing grade measurements between observers.**

further treatment was advised. The implant failure was associated with an average 8.5° increase of the TPA (PostTPA = -0.1° to ReTPA = 8.4°). Subsequent clinical and radiographic follow-up revealed a good outcome. No further treatment was needed for any of these complications. Similarly, the fibular fracture, the patellar fracture, and the tibial crest fracture were incidental findings at radiographic follow-up and no further treatment was needed.

Major postoperative complications included SSI (n = 23) most frequently requiring implant explantation (n = 12), patellar tendinitis (n = 3), late meniscal tear (n = 2), septic arthritis (n = 2), pectineus muscle strain (n = 1), and tibial fracture due to osteomyelitis after implant explantation (n = 1). SSI were treated empirically with antibiotics for 3 to 6 weeks. In case of persistent clinical signs or recurrence, implant explantation was performed, when sufficient

**Table 2. Postoperative complications.**

| Complication type | Complication | Number of cases | Total |
|---|---|---|---|
| Minor | Patellar tendinitis | 2 | 6 (2.4%) |
| | Fibular fracture | 1 | |
| | Implant failure | 1 | |
| | Patellar fracture | 1 | |
| | Tibial crest fracture | 1 | |
| Major | SSI with implant explantation | 12 | 32 (12.9%) |
| | SSI without implant explantation | 11 | |
| | Patellar tendinitis | 3 | |
| | Late meniscal tear | 2 | |
| | Septic arthritis | 2 | |
| | Pectineus m. strain | 1 | |
| | Tibial fracture (osteomyelitis) | 1 | |
| | | | 38 (15.3%) |

bone healing was evident. A sample for bacteriological analysis was performed for each implant explantation. Antibiotics were then adjusted depending on antibiogram results for positive tests. Isolated bacterias were *Staphylococcus pseudintermedius* (n = 4), *Staphylococcus aureus* (n = 2), *Pseudomonas aeruginosa* (n = 1), and *Proteus mirabilis* (n = 1). One case presented further complications, including a proximal metaphyseal tibial fracture caused by osteomyelitis due to multi-resistant (only susceptible to gentamycin) *Proteus mirabilis*. Fracture healing was achieved by open reduction and internal fixation in this case. Most SSI that did not require implant explantation were treated empirically with antibiotics for 1 to 3 weeks. Patellar tendinitis and pectineus muscle strain were treated with cage rest and meloxicam (Metacam; Boehringer Ingelheim™, Reims, France) 0.1 mg/kg orally once daily for 15 days. Late meniscal tears were treated by hemimeniscectomy under arthroscopy. One of these dogs had concurrent implant explantation, which was associated with a sterile bacterial culture. All dogs achieved a good outcome.

In total, 9.3% cases suffered SSI, and 4.8% cases had their implant explanted. SSI were initially suspected at a median of 28 days (±27 days, 7–112 days).

In the control group, 12 (13.5%) cases suffered a SSI, of which 7 (7.9%) had their implant explanted during the first year after the surgery. Bacterial culture was performed in all cases and was positive in 5 cases. Isolated bacterias were *Staphylococcus pseudintermedius* (n = 3), *Enterobacter aerogenes* (n = 1), and *Pseudomonas aeruginosa* (n = 1).

There was no significant difference between the BioMedtrix™ group and the control group for SSI and implant explantation rates (p = 0.31, and p = 0.29, respectively). However, the power of these results is very low (18%). Based on the implant explantation rates (4.8% in the BioMedtrix™ group, 7.9% in the Synthes™ group), a prospective study including 970 patients in each group would have been required to achieve an ideal statistical power of 80%.

## Discussion

In this study, the average ModTPA (1.2˚) and bone healing grade (80% grade 2 or grade 3) were comparable to previous studies using TPLO plates with proximal locking screws (2.4˚, 0.15˚, 1.5˚) [8,9,12]. This supports the mechanical stability of the implant and validated our first and second hypotheses.

Postoperative complications were found in 15.3% of cases. Minor complications represented 2.4% cases, major complications 12.9% cases, and no catastrophic complications were reported, which is comparable to previous recent studies, particularly concerning SSI [7–11]. Also, SSI and implant explantation rates were similar to the control group (p = 0.31 and p = 0.29, respectively). This rejects our third and fourth hypotheses.

To the authors' knowledge, this study is the first to describe the clinical use of the BioMedtrix™ TPLO Curve® plate and the associated long-term outcome.

Changes in postoperative TPA and assessment of bone healing were judged as reliable tools for comparing the mechanical stability of the implant to previous studies [8,9,12]. While ModTPA comparisons were straightforward, different radiographic bone healing assessment methods were used in these studies, making direct comparisons challenging. To overcome this, a modification of the bone healing scale described by Oxley was made to allow numerical grading [12]. In this scale, bone healing grades 2 and 3 correspond to clinical union and bony union, respectively, as previously described [22]. Both grades are considered sufficient for ending exercise restrictions postoperatively. Also, those reports have longer follow-up timeframes (60 to 75 days postoperatively) compared to this study (48 days). As a result, since bone healing grades obtained in this study were comparable to studies with longer mean timeframes, the authors conclude the non-inferiority of the BioMedtrix™ TPLO Curve® plate.

Comparisons were also made for perioperative complications, despite the inconstant use, in previous reports, of the consensual classification developed by Cook *et al.* in 2010 [19]. While the overall intraoperative and postoperative complication rates in this study (6.0% and 15.3%, respectively) were similar to previous recent studies, minor and major complication rates differed [7–11]. This was due to differences in complications classification. Most studies used to classify a complication as minor if a surgical revision wasn't required. In particular, complications treated with medications such as antibiotics were classified as minor, which differs from the consensual classification. After adaptation to classifications used in those studies, minor and major postoperative complication rates in this study were also similar to previous recent studies, with 8.8% and 6.5%, respectively.

Reported intraoperative complications included 6 fibular fractures or osteotomies. This complication has been reported in cases with tibiofibular synostosis or when the proximal fibula is relatively thick [23]. However, Zuckerman *et al.* reported a significant loss of rotation (mean 5.4˚), suggesting an important role of the fibula in mechanical stability. An additional fixation with a caudal plate has been recommended to reduce the loss of rotation [23]. In this study, the authors felt that the shape of the BioMedtrix™ TPLO Curve® plate may have interfered with the caudal placement of a second plate due to its curvature, which could be a potential disadvantage of the plate. Also, no additional stabilization was judged necessary by the surgeons in concerned cases. ModTPA for cases with fibular fracture or osteotomy were 3.7˚ (case n˚17), 1.7˚ (case n˚ 22), 0˚ (case n˚35), 3.5˚ (case n˚111), 0.2˚ (case n˚216), and 2.5˚ (case n˚239). This questions the need for additional fixation when using the BioMedtrix™ TPLO Curve® plate.

The main postoperative complication was SSI. This happened in 9.3% cases, requiring implant explantation in 4.8% cases. SSI were initially suspected at a median of 28 days postoperatively, when the HyProtect™ coating of the BioMedtrix TPLO Curve® plate is still active. Only one dog suffered a SSI suspected after 100 days postoperatively (at day 112). This may question the evidence of efficacy of the plate coating to prevent local infection during this period. Also, a potentially important limitation of this implant system is the absence of coating on the screws. This may significantly impact the antimicrobial effect of the implant system, as infection may develop against screws and away from the coating effect of the plate, for example, due to iatrogenic or hematogenous contaminations. Silver-based coated implants have already been described in human and veterinary medicine, with conflicting results [24–28]. The clinical efficacy of their antimicrobial effect is still debated, mainly because of the lack of well-structured clinical trials, standardization in coating manufacturing, and lack of evidence [27,28]. However, silver, and more precisely its ionic form ($Ag^+$) is known to be an intrinsically antimicrobial material. The mechanism of its bactericidal activity is not completely understood, but inactivation of critical enzymes of the respiration chain by metal binding to thiol groups, and induction of hydroxyl radicals, seems to be an important part of the mechanism [27]. Then, the antimicrobial effect may be limited to tissue in contact with the coated material, and may be ineffective in reducing superficial SSI or SSI located on non-coated screws. To the best of the authors' knowledge, the range of the antimicrobial effect of the coating, and the influence of screw coating when used with a coated plate have not been investigated and warrant further investigation.

In this study, SSI and implant explantation rates (9.3% and 4.8%, respectively) were comparable to previous reports using non-coated implants (6.6% and 2.0%, respectively) [7–11]. However, direct comparison to previous reports to evaluate the effect of the HyProtect™ coating of the BioMedtrix TPLO Curve® plate may be controversial, as SSI may also be influenced by many parameters, including surgical technique, implant (including coating), perioperative care, and antibiotherapy, sometimes with conflicting results [13–15,29–31]. This was mitigated with the inclusion of the control group, formed by a comparable population. The only statistically significant difference between groups was CrCL lesion type (partial or complete rupture),

which is not reported to be a risk factor in SSI development [32]. Between groups, patients management was unchanged between January 2017 and December 2020. In particular, the same surgeons performed the procedures, using similar anesthetic and antibiotic protocols. Comparison to the control group provided similar conclusions for SSI and implant explantation rates. Based on these implant explantation rates (4.8% in the BioMedtrix™ group, 7.9% in the Synthes™ group), a prospective study including 970 patients in each group would have been required to achieve an ideal statistical power demonstrating a 5% difference in implant explantation rates (using $\alpha = 0.05$ and $\beta = 0.8$). This was not achievable in our practice conditions. This lack of statistical power is an important limitation of this study. Similarly, due to the relatively small population presenting SSI, potential risk factors such as breed (German Shepherds) or sex (males) not analyzed [32].

Other limitations are due to the retrospective nature of this study. As such, complications may have been underestimated. In particular, no minor postoperative complication such as superficial wound inflammation or seroma was reported. These complications were reported in previous studies in 3.6% to 16.5% of cases [7–11,33]. Superficial wound inflammations, seroma, or infection that spontaneously resolved may have been underreported because the initial follow-up and stitches removal was mostly performed by the referring veterinarian. This is a limitation in comparison to other previous studies.

In this study, two observers assessed the radiographs. Interobserver variability was evaluated for TPA measurements, and compared to previous studies to evaluate for quality [34–36]. Statistically significant differences were found between observers only for PreTPA measurements (p = 0.028). However, mean interobserver variation (2.10˚) was similar to those studies, ranging from 0.31˚ to 2.4˚, and may be considered clinically not significant [37]. No other significant difference was found between observers. Intraobserver variability evaluations were beyond the scope of this study.

Breed variation of the proximal tibial morphology has been reported [38]. Contouring of the plate may be necessary during the procedure, even with anatomically contoured plates [9]. However, this contouring may increase the risk of intra-articular screw placement [7–9,33]. Otherwise, contouring of the plate with a fixed-angle locking system may require conversion to an unicortical screw or a distally angled bicortical non-locking screw potentially reducing mechanical stability [9]. Placing the plate more distally has also been proposed, but this procedure might necessitate the use of a larger saw blade or the modification of the centering of the osteotomy. Both modifications may lead to complications such as tibial tuberosity fracture or instability at the osteotomy site [39–42]. The BioMedtrix™ TPLO Curve® plate features polyaxial locking plate holes, allowing up to 12.5˚ of screw angulation while maintaining stability. This allows to angle the screw away from the joint. This degree of angulation allowed the use of a bicortical locking screw possible in every case in this study, as we didn't report the implantation of an intra-articular or unicortical screw. The curved shape of the BioMedtrix™ TPLO Curve® plate subjectively allowed a more proximal-caudal implantation on the proximal tibial fragment [33]. This seemed to allow a better bone purchase for screw implantation. Reduction of the oscillating saw diameter and thus the fragment size was also subjectively possible while keeping an appropriate centering of the osteotomy [40–42]. As a result, the tibial tuberosity may present a smaller tibial tuberosity height/tibial width ratio, thus minimizing the risk of tibial tuberosity fracture [39]. Biomechanical studies are necessary to confirm these hypotheses.

## Conclusion

In this study, the BioMedtrix™ TPLO Curve® plate provided a reliable fixation system for osteotomy healing after TPLO procedures in cranial cruciate ligament-deficient dogs, as

ModTPA and radiographic bone healing were similar to previous studies. However, SSI and implant explantation rates were similar to the control group. Complication rates, including SSI rate, were also similar to previous studies. The HyProtect™ coating of the BioMedtrix TPLO Curve® plate, therefore, did not reduce the occurrence of SSI in this study. Further studies are required to evaluate the potential advantages of the antimicrobial effect of the HyProtect® coating.

## Acknowledgments

The authors thank Dr. Marc-Antoine RAPPART (Master 2: Modeling and computing of the living, Faculty of Lyon, France) and for his help in analyzing the data.

## Author Contributions

**Conceptualization:** Iban Irubetagoyena.

**Data curation:** Geoffrey Pagès, Meike Hammer.

**Formal analysis:** Geoffrey Pagès, Meike Hammer, Jean-Guillaume Grand, Iban Irubetagoyena.

**Investigation:** Geoffrey Pagès, Meike Hammer, Iban Irubetagoyena.

**Methodology:** Jean-Guillaume Grand, Iban Irubetagoyena.

**Supervision:** Iban Irubetagoyena.

**Validation:** Jean-Guillaume Grand, Iban Irubetagoyena.

**Writing – original draft:** Geoffrey Pagès, Meike Hammer, Jean-Guillaume Grand, Iban Irubetagoyena.

**Writing – review & editing:** Geoffrey Pagès, Meike Hammer, Jean-Guillaume Grand, Iban Irubetagoyena.

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
