## [Decision Letter · Decision Letter 0]

8 Jun 2022

PONE-D-22-10523Long-term outcome of tibial plateau leveling osteotomy using an antimicrobial silver-based coated plate in dogsPLOS ONE

Dear Dr. Pages,

Thank you for submitting your manuscript to PLOS ONE. After careful consideration, we feel that it has merit but does not fully meet PLOS ONE’s publication criteria as it currently stands. Therefore, we invite you to submit a revised version of the manuscript that addresses the points raised during the review process.

We look forward to receiving your revised manuscript.

Kind regards,

Ashraf M. Abu-Seida, Ph.D.

Academic Editor

PLOS ONE

Journal Requirements:

Additional Editor Comments:

There are few grammatical issues throughout the manuscript, please correct them and follow up the format of the journal

Reviewers' comments:

Reviewer's Responses to Questions

**Comments to the Author**

1. Is the manuscript technically sound, and do the data support the conclusions?

Reviewer #1: Yes

Reviewer #2: Yes

Reviewer #3: Yes

2. Has the statistical analysis been performed appropriately and rigorously? 

Reviewer #1: Yes

Reviewer #2: Yes

Reviewer #3: Yes

3. Have the authors made all data underlying the findings in their manuscript fully available?

Reviewer #1: Yes

Reviewer #2: Yes

Reviewer #3: Yes

4. Is the manuscript presented in an intelligible fashion and written in standard English?

Reviewer #1: Yes

Reviewer #2: Yes

Reviewer #3: Yes

5. Review Comments to the Author

Reviewer #1: The authors have submitted a simple and clean evaluation and are to be commended. I relatively few comments:

line 53 - change preferred to recommended

line 62 - complications should be complication

line 63 - delete "the most"

line 69 - change claiming to designed for

line 72 - delete designed

line 74 - change "to protect" into "as providing protection"

Line 75 - add "of this study" after objective

line 95 - clarify that additional surgery due to complication (explant) was not cause for exclusion. At this early point in the manuscript the point isn't clear.

line 98 (or somewhere before discussion) - the authors need to describe how complications were classified and what exactly constituted a complication. They cover some of this in the discussion but it should be here also. Why is patellar tendinitis listed as both a minor and a major complication?

line 114 - of should be to

line 144 - ModTPA should be explained as ReTPA minus PostTPA (thus the positive values)

line 155 - this might be the place to further describe the application of Cook's system

line 184 - previous values for ModTPA should be reported (probably in the discussion)

Figures 1, 2, 3 - perhaps better positioned after the section discussing the statistical analysis

line 255 change demonstrates t supports

line 308 - efficiency should be evidence of efficacy

OVERALL - despite the brief discussion in lines 315-324, the major deficiency of this study is the lack of a control group relative to this hospital. The data alluded to should be incorporated into this study, or, preferably, a similar cohort population from previous years using a different plate should be evaluated identically. Obviously there is a variable introduced about time, but this data would still be preferable to the historical/literature comparisons currently presented. This will be annoying, but will dramatically improve the manuscript.

Reviewer #2: This paper (PONE-D-22-10523) discusses the outcome including surgical site infection (SSI) after tibial plateau levering osteotomy (TPLO) using the BioMedtrix TPLO Curve®︎ plate. This paper provides the interesting data. However, it is concerned the evidence level of this paper is weak to conclude that the antimicrobial silver-based (HyProtect®︎) coating plate does not reduce SSI, because of no control group, retrospective style, and a single surgeon study at a single institution. In addition, some revisions are necessary to improve the quality of your manuscript as follows.

Major points:

1. Study design (L.94-95): You should describe the exclusion criteria, in more detail (ex. other orthopedic diseases of affected limb, dermatitis, or infections of other area...).

2. Surgical procedure (L.126): Add details about the type and trade name of sterile wound dressings used in this study because of their impact on the occurrence of SSI.

3. Surgical procedure (L.130-131): The postoperative cefalexin dosage seems a little low. Please recheck it.

4. Radiographic evaluation (L.140): The relationship between the person who performed TPLO surgery and the person who evaluated the radiographs is unclear. Please clarify it.

5. Results: Add the percentage (or number) of partial and complete tears of the cranial cruciate ligament.

6. Results (L201-202): Are all intraoperative minor complications the fibula fractures?

7. Results (L216-217) and Table 2: Are patellar fracture and tibial crest fracture minor complication? Please recheck them.

8. Results (L246): Other titles are preferable (ex. Differences in radiographic measurements between observers or Intraobserver variability).

9. Are the screws inserted in these cases coated with the same coating as the TPLO Curve®︎ plates? If not, you need to discuss the effect of non-coating screws on SSI in the Discussion.

10. Discussion (L.305-309): Briefly describe the antimicrobial mechanism of silver-based coated implants.

11. Ref 16 is inappropriate as a reference paper. Please delete it.

Minor points:

1. L.66: 3,6% to 16,5%→3.6% to 16.5%

2. L.166: observers’ mean→observer’s mean?

3. L.333: authors ’opinion→author’s opinion?

4. Figure 1: ”0°” added to vertical axis.

5. Figure 2: ”0°” added to vertical axis.

Reviewer #3: This is a manuscript that investigates a newer implant that offers a special coating which in theory should reduce the incidence of surgical site infection. Such plate coatings are being advertised more aggressively, and their effectiveness should be studied. The results of this smaller scale report indicate a similar infection rate to previous studies and thuse the coating does not appear to offer a clinical benefit.

Overall this manuscript could use some editorial assistance as there are some minor grammatical issues interspersed throughout the manuscript.

Line 53- tibial plateau leveling osteotomy.... not plate

Need to define the minor, major, and catastrophic complications. Referencing the cook paper is appropriate, but those definitions still need to be present within the manuscript

Line 204- cranial tibial artery?

Line 204- the retention of a K-wire is not considered a complication to this reviewer as its presence is not in a direct response to complication such as tibial tuberosity fracture, it was just left in place for perceived increase in construct stability. This is also questionable given reports exist which indicate a retained K wire does not provide significant increase in stability

Line 215-222: Need to better define what is major vs. minor complication. For example, a patella fracture is defined as minor which was likely treated conservatively with rest and NSAIDs. Why is patellar tendinitis classified as a major complication when it would have likely also been treated with rest and NSAIDS. The consistency between these issues is not apparent to this reviewer, again clearly defining the criteria used in this report and reason conditions were categorized as major or minor needs to be clarified

Line 240- where all SSI confirmed by positive culture? If not then what definition and criteria were used. If culture was used for all then consider summary of the bacterial culture results as it would be interesting to see if there is any difference in bacteria isolates with regard to plate coating

Line 315-325- this reviewer is uncertain how to feel about this data being contained in the discussion, if it is from the authors' centers then it could be included in the study design and should be added to the material methods and statistical analysis, essentially changing the study design to a retrospective cohort comparison and potentially strengthening the article. This reviewer would recommend major revision to include this information. If not desired, then it would be better to compare study data results to previous reports (i.e historical data) as previous TPLO reports using non-coated implants are abundant and the study results are comparable to reach the same conclusion. As written, the authors attempt to 'backdoor' data that is not fully explained to strengthen their conclusion. A retrospective cohort study design with direct comparison would be best as it improves the strength of study design.

6. PLOS authors have the option to publish the peer review history of their article (what does this mean?). If published, this will include your full peer review and any attached files.

Reviewer #1: No

Reviewer #2: **Yes: **Kazuya Edamura

Reviewer #3: No

---

## [Author Response · Author response to Decision Letter 0]

12 Jun 2022

Specific comments are also included in the "Response to Reviewers" document.

Academic Editor:

We deposited the protocols on protocols.io as requested. You may find them using the following links:

Anesthetic, analgesic, and antibiotic protocol: dx.doi.org/10.17504/protocols.io.e6nvwkqm2vmk/v1

Radiographic evaluation: dx.doi.org/10.17504/protocols.io.36wgq7qekvk5/v1

Data analysis: dx.doi.org/10.17504/protocols.io.x54v9yowqg3e/v1

However, we are concerned about the peer-reviewed publication of these protocols, if this may extend the duration of the publication process of this article.

The main author is currently very close to the deadline for credentials submission of his ECVS resident formation (31st July 2022).

Additional Editor Comments:

There are few grammatical issues throughout the manuscript, please correct them and follow up the format of the journal.

Corrections have been made throughout the manuscript for identified grammatical issues, thanks to the help of a native English colleague.

\fReviewer #1:

The authors have submitted a simple and clean evaluation and are to be commended. I relatively few comments:

 line 53 - change preferred to recommended: This has been changed 

line 62 - complications should be complication: This has been changed 

line 63 - delete "the most": This has been deleted

 line 69 - change claiming to designed for: This has been changed 

line 72 - delete designed: This has been deleted

 line 74 - change "to protect" into "as providing protection": This has been changed 

Line 75 - add "of this study" after objective: This has been added

 line 95 - clarify that additional surgery due to complication (explant) was not cause for exclusion. At this early point in the manuscript the point isn't clear.: We added to the previous sentence: Additional procedures due to complications (septic arthritis, implant explantation, late meniscal tear…) were not cause for exclusion. 

line 98 (or somewhere before discussion) - the authors need to describe how complications were classified and what exactly constituted a complication. They cover some of this in the discussion but it should be here also. Why is patellar tendinitis listed as both a minor and a major complication?: This has been added in « Complications evaluation ». As a result, patellar tendinitis was classified as a minor complication if no additional treatment was performed and the condition resolved spontaneously, and as a major complication when extended rest and NSAIDs were required to resolve the condition. line 114 - of should be to: This has been changed

 line 144 - ModTPA should be explained as ReTPA minus PostTPA (thus the positive values): This has been added

 line 155 - this might be the place to further describe the application of Cook's system: Indeed, comment for line 98 has been added here

 line 184 - previous values for ModTPA should be reported (probably in the discussion): This has been added in the discussion 

Figures 1, 2, 3 - perhaps better positioned after the section discussing the statistical analysis: Indeed, this was a mistake

 line 255 change demonstrates t supports: This has been changed 

line 308 - efficiency should be evidence of efficacy: This has been changed

 OVERALL - despite the brief discussion in lines 315-324, the major deficiency of this study is the lack of a control group relative to this hospital. The data alluded to should be incorporated into this study, or, preferably, a similar cohort population from previous years using a different plate should be evaluated identically. Obviously there is a variable introduced about time, but this data would still be preferable to the historical/literature comparisons currently presented. This will be annoying, but will dramatically improve the manuscript.: The data have been included in the study. However, a similar cohort population was not achievable, as surgical management and available clinical records were markedly modified before this date.

\fReviewer #2:

This paper (PONE-D-22-10523) discusses the outcome including surgical site infection (SSI) after tibial plateau levering osteotomy (TPLO) using the BioMedtrix TPLO Curve®︎ plate. This paper provides the interesting data. However, it is concerned the evidence level of this paper is weak to conclude that the antimicrobial silver-based (HyProtect®︎) coating plate does not reduce SSI, because of no control group, retrospective style, and a single surgeon study at a single institution. In addition, some revisions are necessary to improve the quality of your manuscript as follows.: In this study, two dipl. ECVS surgeons performed the procedures. This was not accurately explained in the manuscript and has been corrected (please see line 91). Also, a control group was formed for SSI analysis.  

Major points: 

1. Study design (L.94-95): You should describe the exclusion criteria, in more detail (ex. other orthopedic diseases of affected limb, dermatitis, or infections of other area...).: This has been changed to: Dogs with a history of stifle surgery on the affected limb, presented with potential infectious condition (local or generalized dermatitis, prostatitis…), or requiring an additional stifle joint procedure (patellar stabilization…) were excluded. Additional procedures due to complications (septic arthritis, implant explantation, late meniscal tear…) were not cause for exclusion. 

2. Surgical procedure (L.126): Add details about the type and trade name of sterile wound dressings used in this study because of their impact on the occurrence of SSI.: This has been added. 

3. Surgical procedure (L.130-131): The postoperative cefalexin dosage seems a little low. Please recheck it.: Thank you, this was a mistake. The correct used dosage was 20 mg/kg twice a day. 

4. Radiographic evaluation (L.140): The relationship between the person who performed TPLO surgery and the person who evaluated the radiographs is unclear. Please clarify it.: This has been changed to: Two observers (1 Dipl. ECVS and 1 ECVS resident) assessed the anonymized radiographs on a DICOM viewer (Horos™ v3.3.6, horosproject.org). : The Dipl. ECVS observer performed 44% of the procedures.

 5. Results: Add the percentage (or number) of partial and complete tears of the cranial cruciate ligament.: This has been added, along with number of caudal cruciate ligament tears: One hundred and sixty-three (66.0%) dogs presented a complete CrCL rupture, and 84 (34.0%) dogs presented a partial CrCL tear. Fifteen (6.1%) dogs presented a caudal cranial cruciate ligament tear. 

6. Results (L201-202): Are all intraoperative minor complications the fibula fractures?: Yes, we modified the sentence to: All minor intraoperative complications were fibular fractures, which developed during tibial plateau rotation. 

7. Results (L216-217) and Table 2: Are patellar fracture and tibial crest fracture minor complication? Please recheck them.: Yes, as described by Cook and al. (2010), these complications did not require further medical or surgical management, and thus were classified as minor complications. This classification has been detailed in « Complications evaluation ». 

8. Results (L246): Other titles are preferable (ex. Differences in radiographic measurements between observers or Intraobserver variability).: These have been changed 

9. Are the screws inserted in these cases coated with the same coating as the TPLO Curve®︎ plates? If not, you need to discuss the effect of non-coating screws on SSI in the Discussion.: Thanks for this relevant comment. Indeed, coated screws are not provided in this system. The discussion has been modified. 

10. Discussion (L.305-309): Briefly describe the antimicrobial mechanism of silver-based coated implants.): This has been added 

11. Ref 16 is inappropriate as a reference paper. Please delete it.: This reference has been replaced by: Rogatko CP, Warnock JJ, Bobe G et al: Comparison of iatrogenic articular cartilage injury in canine stifle arthroscopy versus medial parapatellar mini-arthrotomy in a cadaveric model. Vet Surg 2018;47:6-14  

Minor points: 

1. L.66: 3,6% to 16,5%→3.6% to 16.5%: This has been corrected 

2. L.166: observers’ mean→observer’s mean?: This has been corrected 

3. L.333: authors ’opinion→author’s opinion?: This has been removed 

4. Figure 1: ”0°” added to vertical axis.: We hope there is no format compatibility issue with the figures. In our version the « 0° » is present for vertical axis. 

5. Figure 2: ”0°” added to vertical axis.: This is the same for Figure 2, we also have « 0° » in our version.

\fReviewer #3:

This is a manuscript that investigates a newer implant that offers a special coating which in theory should reduce the incidence of surgical site infection. Such plate coatings are being advertised more aggressively, and their effectiveness should be studied. The results of this smaller scale report indicate a similar infection rate to previous studies and thuse the coating does not appear to offer a clinical benefit. Overall this manuscript could use some editorial assistance as there are some minor grammatical issues interspersed throughout the manuscript.  

Line 53- tibial plateau leveling osteotomy.... not plate: We apologize for such a mistake in the first lines of the article. This has been corrected. 

Need to define the minor, major, and catastrophic complications. Referencing the cook paper is appropriate, but those definitions still need to be present within the manuscript: This has been added in « Complications evaluation ». 

Line 204- cranial tibial artery?: Indeed. This has been corrected. 

Line 204- the retention of a K-wire is not considered a complication to this reviewer as its presence is not in a direct response to complication such as tibial tuberosity fracture, it was just left in place for perceived increase in construct stability. This is also questionable given reports exist which indicate a retained K wire does not provide significant increase in stability: This has been modified, together with complication rates in the discussion. 

Line 215-222: Need to better define what is major vs. minor complication. For example, a patella fracture is defined as minor which was likely treated conservatively with rest and NSAIDs. Why is patellar tendinitis classified as a major complication when it would have likely also been treated with rest and NSAIDS. The consistency between these issues is not apparent to this reviewer, again clearly defining the criteria used in this report and reason conditions were categorized as major or minor needs to be clarified: Precisions have been added to the description of the complications, both for materials and methods in « Complications evaluation », and in « Postoperative complications » to better explain reasons why specific complications were classified as minor or major. 

Line 240- where all SSI confirmed by positive culture? If not then what definition and criteria were used. If culture was used for all then consider summary of the bacterial culture results as it would be interesting to see if there is any difference in bacteria isolates with regard to plate coating: SSI were diagnosed using the definition proposed by Mangram in 1999 (the reference has been added in the manuscript). As a result, bacterial cultures were infrequently performed for superficial and deep SSI resolved with an empirical antibiotic treatment. Culture was positive in only 8 cases (for a total of 12 implant explantations), sadly preventing analysis, particularly to evaluate a difference related to coating. 

Line 315-325- this reviewer is uncertain how to feel about this data being contained in the discussion, if it is from the authors' centers then it could be included in the study design and should be added to the material methods and statistical analysis, essentially changing the study design to a retrospective cohort comparison and potentially strengthening the article. This reviewer would recommend major revision to include this information. If not desired, then it would be better to compare study data results to previous reports (i.e historical data) as previous TPLO reports using non-coated implants are abundant and the study results are comparable to reach the same conclusion. As written, the authors attempt to 'backdoor' data that is not fully explained to strengthen their conclusion. A retrospective cohort study design with direct comparison would be best as it improves the strength of study design.: A control group has been formed as suggested.

---

## [Decision Letter · Decision Letter 1]

30 Jun 2022

PONE-D-22-10523R1Long-term outcome of tibial plateau leveling osteotomy using an antimicrobial silver-based coated plate in dogsPLOS ONE

Dear Dr. Pages,

Thank you for submitting your manuscript to PLOS ONE. After careful consideration, we feel that it has merit but does not fully meet PLOS ONE’s publication criteria as it currently stands. Therefore, we invite you to submit a revised version of the manuscript that addresses the points raised during the review process.

We look forward to receiving your revised manuscript.

Kind regards,

Ashraf M. Abu-Seida, Ph.D.

Academic Editor

PLOS ONE

Journal Requirements:

Additional Editor Comments (if provided):

Please follow up the format of the journal

Please respond to all reviewers comments

Reviewers' comments:

Reviewer's Responses to Questions

**Comments to the Author**

1. If the authors have adequately addressed your comments raised in a previous round of review and you feel that this manuscript is now acceptable for publication, you may indicate that here to bypass the “Comments to the Author” section, enter your conflict of interest statement in the “Confidential to Editor” section, and submit your "Accept" recommendation.

Reviewer #1: All comments have been addressed

Reviewer #2: (No Response)

Reviewer #3: (No Response)

2. Is the manuscript technically sound, and do the data support the conclusions?

Reviewer #1: Yes

Reviewer #2: Yes

Reviewer #3: Yes

3. Has the statistical analysis been performed appropriately and rigorously? 

Reviewer #1: Yes

Reviewer #2: Yes

Reviewer #3: Yes

4. Have the authors made all data underlying the findings in their manuscript fully available?

Reviewer #1: Yes

Reviewer #2: Yes

Reviewer #3: Yes

5. Is the manuscript presented in an intelligible fashion and written in standard English?

Reviewer #1: Yes

Reviewer #2: Yes

Reviewer #3: Yes

6. Review Comments to the Author

Reviewer #1: The authors have addressed my concerns. I maintain (i assume they agree) that a better control population would be far superior, but i accept that is not going to happen. In recognition of the time concern, i recommend publication

Reviewer #2: This manuscript has been appropriately revised. Before completing the manuscript, please recheck the following minor points.

1. You should include the numbers in the BioMedtrix TPLO Curved plate and Synthes TPLO plate groups in the Abstract.

2. Abstract (Line 39): 7→6?, 31→32? Please final check them (and also others)!

3. Abstract (Results): Please describe which group results.

4. Line 159: “ModTPA was calculated (ReTPA minus PostTPA)”→”ReTPA minus PostTPA (ModTPA) was then calculated”

5. Results (Line: 210-218): Each data such as age, breed, body weight, affected stifle joint, type of rupture etc. should be listed separately for each group.

6. I found that the cases with combined caudal cruciate ligament rupture was included in this revised manuscript. Did you perform only TPLO surgery in these cases? Or did you perform a combination of extracapsular stabilization technique, etc.?

7. Line 261-262: Names of bacteria should be in italics.

8. Line 282: Another title is preferable (from “Statistical analysis” to another title). You did not change it, although you described “These have been changed” in revision letter.

9. Line 392: Others limitations→Other limitations?

10. Figures: “0°” on the vertical axis has not been improved...

Reviewer #3: The authors have revised this manuscript and while improved still needs to better incorporate the control group with thorough analysis and better description of analysis in the materials in methods.

Line 96-100: add the information from lines 109-113 to this region of the paragraph

Data analysis: need to add statistical methods for comparison of the population statistics between the biomedtrix and synthes groups for continuous and categorical data, also need to add post hoc power analysis performed

Line 218-219 : need to expand information of the synthes group similar to the biomedtrix group, add median values and ranges for data of the synthes group and statistically compare between groups using the methodology in the data analysis section. must ensure there is not a bias in one group that would predispose to SSI

Line 288-290: add in the post hoc power analysis information contained in the discussion at lines 385-388. this is data

7. PLOS authors have the option to publish the peer review history of their article (what does this mean?). If published, this will include your full peer review and any attached files.

Reviewer #1: No

Reviewer #2: **Yes: **Kazuya Edamura

Reviewer #3: No

---

## [Author Response · Author response to Decision Letter 1]

3 Jul 2022

Specific comments are also included in the "Response to Reviewers" document.

Journal Requirements:

Please review your reference list to ensure that it is complete and correct. If you have cited papers that have been retracted, please include the rationale for doing so in the manuscript text, or remove these references and replace them with relevant current references. Any changes to the reference list should be mentioned in the rebuttal letter that accompanies your revised manuscript. If you need to cite a retracted article, indicate the article’s retracted status in the References list and also include a citation and full reference for the retraction notice. : Thank you for the comment. One reference (n°16: Radasch RM: Cruciate surgery options: the good, the bad and the ugly. In: The North American Veterinary Conference; January 20, 2015; Marriott Hotel, Orlando, FL) has been removed previously (as requested by Reviewer #2), and has been replaced with: Rogatko CP, Warnock JJ, Bobe G et al: Comparison of iatrogenic articular cartilage injury in canine stifle arthroscopy versus medial parapatellar mini-arthrotomy in a cadaveric model. Vet Surg 2018;47:6-14. Also, a new reference has been added as N°20: Mangram AJ, Horan TC, Pearson ML et al: Guideline for Prevention of Surgical Site Infection, 1999. Am J Infect Control 1999;27:97-132. The following references has been shifted.

Additional Editor Comments (if provided):

Please follow up the format of the journal : To the best of the authors’ knowledge, the submitted manuscript follows the format of the journal.

Please respond to all reviewers comments : Corrections have been performed, you may find them below.

Reviewer #1:

The authors have addressed my concerns. I maintain (i assume they agree) that a better control population would be far superior, but i accept that is not going to happen. In recognition of the time concern, i recommend publication : Thank you. Indeed, we are sadly unable to provide comparable data concerning complications in the control group, and include additional cases, due to protocol inconstancies previous to the inclusion starting date (January 2017).

Reviewer #2:

This manuscript has been appropriately revised. Before completing the manuscript, please recheck the following minor points. 

1. You should include the numbers in the BioMedtrix TPLO Curved plate and Synthes TPLO plate groups in the Abstract. : This has been corrected

 2. Abstract (Line 39): 7→6?, 31→32? Please final check them (and also others)! : This has been corrected. Others have been checked as well and some mistakes have been corrected in the beginning of the discussion in particular 

3. Abstract (Results): Please describe which group results. : This has been done 

4. Line 159: “ModTPA was calculated (ReTPA minus PostTPA)”→”ReTPA minus PostTPA (ModTPA) was then calculated” : This has been corrected 

5. Results (Line: 210-218): Each data such as age, breed, body weight, affected stifle joint, type of rupture etc. should be listed separately for each group. : This has been added. 

6. I found that the cases with combined caudal cruciate ligament rupture was included in this revised manuscript. Did you perform only TPLO surgery in these cases? Or did you perform a combination of extracapsular stabilization technique, etc.? : Indeed, these cases all presented partial tears (the term « partial » has been added in this revised manuscript), and were treated with only TPLO. 

7. Line 261-262: Names of bacteria should be in italics. : This has been corrected, and names of bacterias have been added for the control group 

8. Line 282: Another title is preferable (from “Statistical analysis” to another title). You did not change it, although you described “These have been changed” in revision letter. : Sorry, we may have misunderstood your request. We thought the issue was the title of the figures. The paragraph has been deleted, and sentences have been included in the corresponding paragraphs above for better comprehension of the analysis. 

9. Line 392: Others limitations→Other limitations? : This has been corrected 

10. Figures: “0°” on the vertical axis has not been improved… : This has been corrected converting the files into .tiff images (uploaded separately). We hope the visualization will be correct this time.

Reviewer #3:

The authors have revised this manuscript and while improved still needs to better incorporate the control group with thorough analysis and better description of analysis in the materials in methods.  

Line 96-100: add the information from lines 109-113 to this region of the paragraph : This has been done 

Data analysis: need to add statistical methods for comparison of the population statistics between the biomedtrix and synthes groups for continuous and categorical data, also need to add post hoc power analysis performed : These have been added. Also, this paragraph has been modified to simplify explanations for better comprehension 

Line 218-219 : need to expand information of the synthes group similar to the biomedtrix group, add median values and ranges for data of the synthes group and statistically compare between groups using the methodology in the data analysis section. must ensure there is not a bias in one group that would predispose to SSI : This has been done. Also, averages values were modified to median values in the BioMedtrix group in this paragraph and throughout the manuscript. Bias evaluation was commented in the discussion : This was mitigated with the inclusion of the control group, formed by a comparable population. The only statistically significant difference between groups was CrCL lesion type (partial or complete rupture), which is not reported to be a risk factor in SSI development [32]. 

Line 288-290: add in the post hoc power analysis information contained in the discussion at lines 385-388. this is data : This has been done

---

## [Decision Letter · Decision Letter 2]

19 Jul 2022

PONE-D-22-10523R2Long-term outcome of tibial plateau leveling osteotomy using an antimicrobial silver-based coated plate in dogsPLOS ONE

Dear Dr. Pages

Thank you for submitting your manuscript to PLOS ONE. After careful consideration, we feel that it has merit but does not fully meet PLOS ONE’s publication criteria as it currently stands. Therefore, we invite you to submit a revised version of the manuscript that addresses the points raised during the review process.

Please submit your revised manuscript by  Sep 02 2022 11:59PM. If you will need more time than this to complete your revisions, please reply to this message or contact the journal office at plosone@plos.org. Please include the following items when submitting your revised manuscript:A rebuttal letter that responds to each point raised by the academic editor and reviewer(s). You should upload this letter as a separate file labeled 'Response to Reviewers'.A marked-up copy of your manuscript that highlights changes made to the original version. You should upload this as a separate file labeled 'Revised Manuscript with Track Changes'.An unmarked version of your revised paper without tracked changes. You should upload this as a separate file labeled 'Manuscript'.If applicable, we recommend that you deposit your laboratory protocols in protocols.io to enhance the reproducibility of your results. Protocols.io assigns your protocol its own identifier (DOI) so that it can be cited independently in the future. For instructions see: https://journals.plos.org/plosone/s/submission-guidelines#loc-laboratory-protocols. Additionally, PLOS ONE offers an option for publishing peer-reviewed Lab Protocol articles, which describe protocols hosted on protocols.io. Read more information on sharing protocols at https://plos.org/protocols?utm_medium=editorial-email&utm_source=authorletters&utm_campaign=protocols.

We look forward to receiving your revised manuscript.

Kind regards,

Ashraf M. Abu-Seida, Ph.D.

Academic Editor

PLOS ONE

Journal Requirements:

Additional Editor Comments:

Line 234-235: caudal cruciate ligament tear→caudal cruciate ligament partial? tear

Line 260-267: The number of major intraoperative complications is 11 (4.4%) or 9 (3.6%)? Which is correct?

Table 2: Sort the major complications in order of number of occurrences.

Reviewers' comments:

**Comments to the Author**

1. If the authors have adequately addressed your comments raised in a previous round of review and you feel that this manuscript is now acceptable for publication, you may indicate that here to bypass the “Comments to the Author” section, enter your conflict of interest statement in the “Confidential to Editor” section, and submit your "Accept" recommendation.

Reviewer #2: All comments have been addressed

2. Is the manuscript technically sound, and do the data support the conclusions?

Reviewer #2: Yes

3. Has the statistical analysis been performed appropriately and rigorously? 

Reviewer #2: Yes

4. Have the authors made all data underlying the findings in their manuscript fully available?

Reviewer #2: Yes

5. Is the manuscript presented in an intelligible fashion and written in standard English?

Reviewer #2: Yes

6. Review Comments to the Author

Reviewer #2: This submitted paper (PONE-D-22-10523R2) are appropriately revised in response to the reviewer's remarks. I think that this paper is worthy of acceptance in the PLOS ONE. When you send final manuscript, please recheck the following points.

Line 234-235: caudal cruciate ligament tear→caudal cruciate ligament partial? tear

Line 260-267: The number of major intraoperative complications is 11 (4.4%) or 9 (3.6%)? Which is correct?

Table 2: Sort the major complications in order of number of occurrences.

7. PLOS authors have the option to publish the peer review history of their article (what does this mean?). If published, this will include your full peer review and any attached files.

Reviewer #2: **Yes: **Kazuya Edamura

---

## [Author Response · Author response to Decision Letter 2]

19 Jul 2022

Review comments to the author:

Reviewer #2:

This submitted paper (PONE-D-22-10523R2) are appropriately revised in response to the reviewer's remarks. I think that this paper is worthy of acceptance in the PLOS ONE. When you send final manuscript, please recheck the following points.

Line 234-235: caudal cruciate ligament tear→caudal cruciate ligament partial? tear: This has been corrected

Line 260-267: The number of major intraoperative complications is 11 (4.4%) or 9 (3.6%)? Which is correct?: This has been corrected to 9 (3.6%). Also overall intraoperative complications number has been corrected from 17 (6.9%) to 15 (6.0%). This mistake was due to the complications initially considered as major by the authors (osteotomy stabilizing Kirschner wire left in place to maintain adequate stabilization (n=2)), and removed during the revision process. Thank you for highlighting this mistake.

Table 2: Sort the major complications in order of number of occurrences.: This has been corrected

---

## [Editor Report · Decision Letter 3]

22 Jul 2022

Long-term outcome of tibial plateau leveling osteotomy using an antimicrobial silver-based coated plate in dogs

PONE-D-22-10523R3

Dear Dr. Pages

We’re pleased to inform you that your manuscript has been judged scientifically suitable for publication and will be formally accepted for publication once it meets all outstanding technical requirements.

Kind regards,

Ashraf M. Abu-Seida, Ph.D.

Academic Editor

PLOS ONE
---

## [Editor Report · Acceptance letter]

4 Aug 2022

PONE-D-22-10523R3 

Long-term outcome of tibial plateau leveling osteotomy using an antimicrobial silver-based coated plate in dogs 

Dear Dr. Pagès:

I'm pleased to inform you that your manuscript has been deemed suitable for publication in PLOS ONE. Congratulations! Your manuscript is now with our production department. 

Kind regards, 

on behalf of

Professor Ashraf M. Abu-Seida 

Academic Editor

PLOS ONE